# Identification of the *BcLEA* Gene Family and Functional Analysis of the *BcLEA73* Gene in Wucai (*Brassica campestris* L.)

**DOI:** 10.3390/genes14020415

**Published:** 2023-02-05

**Authors:** Yueyue Jiang, Shengnan Zhang, Hongcheng Xu, Hong Tian, Mengyun Zhang, Shidong Zhu, Chenggang Wang, Jinfeng Hou, Guohu Chen, Xiaoyan Tang, Wenjie Wang, Jianqiang Wu, Xingxue Huang, Jinlong Zhang, Lingyun Yuan

**Affiliations:** 1Vegetable Genetics and Breeding Laboratory, College of Horticulture, Anhui Agricultural University, Hefei 230036, China; 2Anhui Provincial Engineering Laboratory of Horticultural Crop Breeding, Hefei 230036, China; 3Department of Vegetable Culture and Breeding, Wanjiang Vegetable Industrial Technology Institute, Maanshan 238200, China; 4Vegetable Industry Office, Agricultural and Rural Bureau of He County, Maanshan 238201, China

**Keywords:** bioinformatics analysis, functional analysis, gene cloning, LEA protein, Wucai

## Abstract

Late embryogenesis abundant (LEA) proteins are important developmental proteins in the response of plants to abiotic stress. In our previous study, *BcLEA73* was differentially expressed under low-temperature stress. Herein, we combined bioinformatics analysis, subcellular localization, expression assays, and stress experiments (including salt, drought, and osmotic stress) to identify and analyze the BcLEA gene family. Gene cloning and functional analysis of *BcLEA73* were performed in tobacco and *Arabidopsis.* Based on the sequence homology and the available conservative motif, 82 *BrLEA* gene family members were identified and were divided into eight subfamilies in the genome-wide database of Chinese cabbage. The analysis showed that the *BrLEA73* gene was located on chromosome A09 and belonged to the LEA_6 subfamily. Quantitative real-time PCR analysis indicated that the *BcLEA* genes were differentially expressed to varying degrees in the roots, stems, leaves, and petioles of Wucai. The overexpressed *BcLEA73* transgenic plants exhibited no significant differences in root length and seed germination rates compared to the wild-type (WT) plants under control conditions. Under salt and osmotic stress treatment, the root length and seed germination rates of the *BcLEA73-OE* strain were significantly greater than those of WT plants. Under salt stress, the total antioxidant capacity (T-AOC) of the *BcLEA73-OE* lines increased significantly, and the relative conductivity, (REL), hydrogen peroxide (H_2_O_2_) content, and superoxide anion (O_2_^−^) production rate decreased significantly. Under drought treatment, the survival rate of the *BcLEA73-OE* lines was significantly higher than that of WT plants. These results showed that the *BcLEA73* gene of Wucai functions in enhancing the tolerance of plants to salt, drought, and osmotic stress. This study provides a theoretical basis to explore the relevant functions of the *BcLEA* gene family members of Wucai.

## 1. Introduction

Late embryogenesis abundant (LEA) proteins are small molecule-specific peptides generated in the late seed developmental period and were first identified from cotton cotyledons. They are named as such due to their substantial accumulation in the late embryo stage [1]. In most plants, LEA family proteins consist of a large number of polypeptides with high diversity, and they have been classified as a specific subfamily according to various classification features in several published reports. They play vital roles in helping plants to tolerate abiotic stresses, thereby enabling normal life activities.

LEA proteins possess tissue-specific expression. They can be extensively expressed in the roots, stems, leaves, and petioles of plants during the late embryonic development stage of the seeds, and varied expression levels have been detected in response to different abiotic stresses. Although the mechanistic functions of LEA proteins remain unclear, they are involved in cellular adaptation to adversity, such as drought and chilling injury, and they act as chaperones in protecting cellular proteomes and membrane structures [2,3]. A greater expression of LEA proteins is essential for plants to maintain normal life activities during growth or under abiotic stress-related processes. It has been shown that LEA genes are upregulated under abscisic acid (ABA) treatment, suggesting that they are associated with ABA-related pathways [4,5,6].

Wucai is a biennial herb widely grown in autumn and winter in the Yangtze and Huaihe River basins. It is popular among consumers for its richness in Vc and other nutrients. Unfavorable environmental conditions, such as droughts, low temperatures, and salt concentrations, inevitably occur throughout the growth of Wucai, and these abiotic stresses cause varying degrees of injury. Plants have evolved a complex set of resistance mechanisms to cope with adversity stress, including a large class of functional proteins that are directly involved in stress. Genome-wide identification effectively reveals the phylogenetic relationships and ancestral gene structures of plants. Genome-wide identification provides an important information base for the genetic resolution of important agronomic and quality traits. In this study, the *BcLEA* gene family members were identified within a genome-wide context, and their related functions were analyzed to provide a theoretical basis for developing new stress-resistant varieties of Wucai.

## 2. Materials and Methods

### 2.1. Plant Materials

Two Wucai genotypes, namely, ‘Wu 18’ and ‘WS-1′, as well as one wild-type *Arabidopsis* (Col-0 (WT)), were used in this experiment, and they were provided by the Vegetable Genetics and Breeding Team at the College of Horticulture, Anhui Agricultural University.

The seedlings were planted in a greenhouse at 25 ± 2 °C (during the day) and 15 ± 2 °C (during the night) with 70–75% relative humidity and 300 µmol·m^−2^·s^−1^ photon flux density. When the seedlings grew four to five leaves, they were treated at room temperature (NT) at 25 °C/18 °C (day/night) and at low temperature (LT) of 9 °C/4 °C (day/night), with 300 µmol·m^−2^·s^−1^ photon flux density for three days. Then, the functional leaves of the treatment and control groups were snap-frozen and stored at −80 °C for transcriptome sequencing and subsequent experiments. Leaves were sampled at 0 h, 12 h, 24 h, 48 h, and 72 h after low-temperature treatment for quantitative real-time PCR (qRT-PCR) expression analysis. All of the samples had three biological repetitions.

### 2.2. Identification of the BrLEA Gene Family Members and Analysis of Physicochemical Properties

The whole-genome data of Chinese cabbage and *Arabidopsis* were downloaded from the plant genome database “http://brassicadb.cn (accessed on 6 July 2021)” [7]. The HMM configuration files of the eight subfamilies of the LEA gene family were downloaded from Pfam [8] “(http://pfam.sanger.ac.uk/):PF03760(LEA_1) (accessed on 6 July 2021)”, and they included PF03168(LEA_2), PF03242(LEA_3), PF02987(LEA_4), PF00477(LEA_5), PF04927(LEA_6), PF10714(SMP), and PF00257(Dehydrin). A hidden Markov model (HMM) was used to analyze and screen out possible *BrLEA* gene family members. The SMART [9] “http://smart.emblheidelberg.de/ (accessed on 6 July 2021)” and NCBI-CDD “https://www.ncbi.nlm.nih.gov/Structure/bwrpsb/bwrpsb.cgi (accessed on 6 July 2021)” databases were used to analyze alternative *BrLEA* protein sequences. The Expasy website “https://web.expasy.org/protparam/ (accessed on 6 July 2021)” was used to predict the physicochemical properties of the *BrLEA* gene family members [10].

### 2.3. Analysis of Conserved Motifs and Gene Structure of the BrLEA Gene Family Members

The Multiple Expectation Maximization (MEME) program “http://meme-suite.org/tools/meme (accessed on 8 July 2021)” was used to predict the conserved motifs [11]. The exon–intron and conserved motif gene structure of the *BrLEA* gene family members were visualized through TBtools.

### 2.4. Chromosome Location Analysis of the BrLEA Gene FAMILY

The location information of the chromosomes was obtained from the Ensembl plant database “http://plants.ensembl.org/index.html (accessed on 8 July 2021)”, and TBtools was used for visualization.

### 2.5. Cis-Acting Element Analysis of the BrLEA Gene Family

The PlantCARE “http://bioinformatics.psb.ugent.be/webtools/plantcare/htmL/ (accessed on 9 July 2021)” database was used to predict and analyze *cis*-acting elements [12], and the selected promoter *cis-*acting element data were visualized with TBtools.

### 2.6. Screening of Differentially Expressed BcLEAs Based on RNA-Seq

Differentially expressed genes were screened based on the existing low-temperature treatment transcriptome data (accession number: PRJNA735896), which compared the fragments per kilobase of exons per million mapped fragments (FPKM) of differentially expressed *BcLEA* genes under low-temperature stress [13]. TBtools was used to visualize the value, and the results are presented in the form of a heatmap.

The raw RNA-Seq datasets were available from the Sequence Read Archive of the National Center for Biotechnology Information “https://dataview.ncbi.nlm.nih.gov/object/PRJNA735896?reviewer=fnp3ih4ojrq5ohq5vmstrr5koe (accessed on 6 February 2022)” [14]. Total RNA was isolated from the leaves treated at 4 °C for 0, 6, 12, and 24 h, including three biological repeats for each condition. RNA-Seq was performed by the Beijing Genomics Institute (Shenzhen, China).

### 2.7. Analysis of the BcLEA Genes Expression Patterns in Wucai

The specific fluorescent quantitative PCR primers were designed by Oligo 7 and submitted to General Biosystems (Chuzhou, China) for synthesis. *BcActin* was used as the internal reference gene for Wucai. The fluorescent primer design of each gene is shown in Table 1.

The existing transcriptome data for the low-temperature stress treatment from our earlier analysis showed that the expression level of *BcLEA* genes differed significantly under low temperatures. We selected the *BcLEAs* with significant expression levels for verification, and we explored the expression levels of *BcLEA* genes in various tissues under low-temperature stress at different times using qRT-PCR.

### 2.8. Cloning of the BcLEA73 Gene of Wucai

We analyzed the relevant sequences of the *BrLEA73* gene, designed specific primers with primer synthesis Premier 5.0, and sent the primers to the General Biology Company for synthesis. The specific primer sequences are shown in Table 2.

The cDNA synthesized by reverse transcription was used as a template to amplify the target fragment, and the amplified product was detected by agarose gel electrophoresis, and, following this, a gel extraction kit was used to isolate specific gene fragments.

### 2.9. Subcellular Localization Analysis of the BcLEA73 Gene in Wucai

We merged the full-length open reading frame (ORF) of *BcLEA73* and green fluorescent protein (GFP) into the pCAMBIA2300-35S::GFP vector. The experimental method and operation steps for subcellular localization were consistent with the construction of the overexpression vector. The constructed vector was transferred to *Escherichia coli,* and the bacteria were preserved. Tobacco was selected as the infection material for subcellular localization.

### 2.10. Genetic Transformation Analysis of the BcLEA73 Gene in Arabidopsis

We transferred 2 mL of the resuscitated pCAMBIA1305-35s::*BcLEA73*-nFLAG overexpression vector *Agrobacterium tumefaciens* into 200 mL Luria-Bertani (LB) liquid medium. When the OD_560_ of the bacterial solution was 1-2 (the LB liquid medium containing Kan and Rif was zeroed), the solution was centrifuged at 4 °C and 8000 r/min for 5 min, and the supernatant was discarded. This process was repeated two to three times. When the OD_560_ was approximately 0.8 (the buffer was zeroed), the supernatant was transfected into *A. thaliana*. The infected plants were placed in the dark for 24 h and then cultured under normal light, and, following this, sexual plant screening was performed once the seeds had matured.

### 2.11. Analysis of Abiotic Stress on Seed Germination Rate and Root Length

Osmotic stress treatment consisted of the following. Transgenic *Arabidopsis* and wild-type *Arabidopsis* seeds were sown on a ½ Murashige & Skoog (MS) Petri dish containing 250 mM of mannitol.

Salt stress treatment consisted of the following. Sterilized transgenic *Arabidopsis* and wild-type *Arabidopsis* seeds were sown on a ½ MS Petri dish containing 150 mM of NaCl.

Low-temperature treatment consisted of the following. Sterilized transgenic *Arabidopsis* and wild-type *Arabidopsis* seeds were sown on a ½ MS Petri dish and treated at a low temperature of 4 °C.

The control group was sown in ½ MS medium. The control group was observed and recorded every day, the germination rate of the seeds was counted for seven days, and the root length of the *Arabidopsis* seedlings was measured after two weeks.

### 2.12. Measurement of O_2_^−^ Production Rate, H_2_O_2_ Content, Relative Electrolyte Leakage (REL), and Total Antioxidant Capacity (T-AOC)

Determination of relative conductivity consisted of the following. The roots of the *Arabidopsis* seedlings were removed, and the upper parts were placed in a tube containing 20 mL of deionized water and then placed in a thermostatic water bath at 25 °C for 30 min. Following this, the conductivity (L1) of water was measured using a Thermo OrionSTARA HB conductivity meter (Thermo Orion., Waltham, MA, USA). Twenty milliliters of deionized water were added into the blank control tube, and the conductivity (L0) of water was measured. The tube was placed in a thermostatic water bath at 100 °C for 10 min and then cooled to room temperature, and the conductivity (L2) was measured again. The relative conductivity leakage was calculated using the equation REL = (L1 − L0)/(L2 − L0) × 100%. The O_2_^−^ production rate, H_2_O_2_ content, and total antioxidant capacity (T-AOC) were measured using reagent kits (Cat#BC1290, Cat#BC3590, and Cat#BC1310; Beijing Solarbio Science & Technology Co., Ltd., Beijing, China).

## 3. Results

### 3.1. Bioinformatics Analysis of the BrLEA Gene Family

#### 3.1.1. Identification of the BrLEA Gene Family Members and Analysis of Physicochemical Properties

A total of 82 *BrLEA* gene family members were identified based on the genome of Chinese cabbage (Appendix A). The grand average of hydropathicity (GRAVY) of the 82 members ranged from 0.152 to −1.704. Most proteins showed great hydrophilicity, and the GRAVY of 17 proteins exceeded 0, indicating that they were hydrophobic. The amino acid sequences of the 82 *BrLEA* gene family members were 57–777 aa in length, 0.66–84.31 kD in molecular weight, and had isoelectric points of 4.5–10.45. There were 35 family members whose isoelectric points were less than 7.0; 45 family members had isoelectric points greater than 7.5; and two family members had isoelectric points between 7 and 7.5. Subcellular localization prediction of the *BrLEA* gene family members showed that seven members were predicted to be located on the cell membrane, six were predicted to be located in the cell wall, five were predicted to be located in the cytoplasm, 30 were predicted to be located in the chloroplast, 33 were predicted to be located in the nucleus, and only one was predicted to be located in the mitochondria.

#### 3.1.2. Construction of the Gene Family Evolutionary Tree

To explore the classification of the BrLEA gene family, a phylogenetic tree was constructed in relation to 51 Arabidopsis *AtLEA* [15] and 82 *BrLEA* gene family members in Chinese cabbage (Figure 1). The results showed that the *AtLEA* and *BrLEA* family members were divided into eight subfamilies. Among the 82 members of the *BrLEA* gene family, 37 were LEA_2 subfamily members, representing the highest subfamily members. The least numerous were LEA_5 and LEA_6, each with only three members.

#### 3.1.3. Analysis of the Conserved Motifs and Gene Structure of the Family Members

Since the 82 *BrLEA* genes were not highly similar, the protein sequences of each subfamily gene were submitted to MEME for domain or motif structure analysis (Figure 2). The motifs in each subfamily encoded a conserved LEA gene domain. The most closely related genes in each subfamily showed a similar motif composition, indicating that genes within the *BrLEA* subfamily had similar functions.

Analysis of the gene structure of *BrLEA* family members showed that 40 genes of the 82 family members had one intron (Figure 3). In addition, there were 28 genes with exons and no introns, 10 genes with two introns, and two genes with three introns in the 82 family members. Importantly, all LEA_ 5 subfamilies had one intron, and all LEA_ 6 subfamilies had no intron.

#### 3.1.4. Chromosome Location Analysis

Eighty genes in the 82 *BrLEA* genes identified were unequally distributed on 10 chromosomes (Figure 4). The number of genes on both chromosomes A07 and A10 was five, which was the lowest. There were 12 genes on both chromosomes A03 and A04, representing the greatest number. *BrLEA81* and *BrLEA82* were not distributed on chromosomes. In addition, the *BrLEA* gene had three pairs of tandemly duplicated genes on the chromosome.

#### 3.1.5. *Cis*-Acting Element Analysis

There were multiple response hormones and abiotic stress-related *cis*-acting elements on the promoter region of the *BrLEA* family members (Figure 5), indicating that the expression of the *BrLEA* gene was influenced by many factors. This study mainly analyzed the elements related to stress. Abscisic acid response element (ABRE), dehydration response element (DRE), low-temperature response element (LTRE), and other elements might play an important role in the process of plant resistance to abiotic stress.

### 3.2. Analysis of the BcLEA Gene Transcriptome Data of Wucai

Based on the heatmap analysis (Figure 6), we found that the expression levels of the *BcLEA23*, *BcLEA46,* and *BcLEA36* genes showed a downward trend following low-temperature treatment. The expression levels of the *BcLEA32*, *BcLEA49*, *BcLEA55*, *BcLEA73,* and *BcLEA78* genes were upregulated to varying degrees, and the expression level of *BcLEA73* was the largest.

### 3.3. Analysis of BcLEA Gene Expression Patterns

Quantitative real-time PCR was used to investigate the tissue-specific expression of *BcLEAs* in the roots, leaves, stems, and petioles. The genes with the highest expression in the leaves were *BcLEA31*, *BcLEA40*, *BcLEA46,* and *BcLEA73*, of which *BcLEA73* had the greatest expression in the leaves (Figure 7A–K). Highly expressed genes in the stems included *BcLEA55*, *BcLEA19*, *BcLEA69*, *BcLEA77*, and *BcLEA80*, of which *BcLEA55* exhibited the greatest expression (Figure 7A–K). The expression of *BcLEA55* was also the highest in the petioles (Figure 7A–K). Following low-temperature treatment, the expression of *BcLEA73* in the stems, leaves, and petioles showed an upward trend, particularly in the petioles (Figure 7L).

### 3.4. Analysis of BcLEA Gene Expression under Low-Temperature Treatment

To further determine the role of *BcLEA73* under low-temperature treatment, we analyzed the expression level. The expression level of the *BcLEA* gene first decreased after 12 h of low-temperature treatment, and, following this, the expression level showed an upward trend with the extension of treatment time and was highest after 72 h of low-temperature treatment (Figure 8A–E). The expression of *BcLEA78* was the highest after 48 h of low-temperature treatment (Figure 8E). The test results are consistent with the transcriptome data.

### 3.5. Subcellular Localization Analysis of BcLEA73 in Wucai

The infiltrated tobacco leaves were processed onto slides and then observed and photographed under a laser scanning confocal microscope. The results showed that there were green fluorescence signals in the cell membrane and nucleus, indicating that BcLEA73 was expressed in the cell membrane and nucleus (Figure 9). This differs from the results predicted by subcellular localization, probably due to the diversity of LEA genes.

### 3.6. Overexpression of BcLEA73 in Arabidopsis Increases Abiotic Stress Tolerance

To further determine the role of *BcLEA73* in other abiotic stresses, *BcLEA73*-overexpressing lines and WT plants were subjected to osmotic and salt treatments.

We selected leaves of T2 transgenic *Arabidopsis* showing good growth, extracted RNA according to the corresponding kit instructions, and then reverse-transcribed them into cDNA. The relative expression of each strain was obtained through qRT-PCR. We found that the relative expression levels of *BcLEA73* in the OE#1, OE#2, and OE#3 lines were relatively high (Figure 10). Thus, the three overexpression lines were selected for subsequent functional analysis.

### 3.7. Effects of Various Stress Treatments on Seed Germination Rate

Under normal conditions, there was no significant difference in the seed germination rate between the WT and *BcLEA73-OE* strains, and the germination rate reached 100% on the fifth day (Figure 11A). The seed germination rate of *BcLEA73-OE* was significantly higher than that of WT under salt stress treatment (Figure 11B). After 250 mM mannitol treatment, the seed germination rate of WT was significantly lower than that of *BcLEA73-OE* after two days of growth, but there were no differences in seed germination rate among the transgenic lines (Figure 11C). In conclusion, the investigation showed that the germination rate of *BcLEA73-OE* was significantly greater than that of WT. *BcLEA73* played a regulatory role in osmotic and salt stress in the seeds and improved their germination ability in response to stress treatment.

### 3.8. Effects of Different Stress Treatments on Root Length

After 10 days of culture in regular medium, there were no significant differences in the *Arabidopsis* seedlings in terms of root length between WT and *BcLEA73-OE* (Figure 12 A,E). We measured the root length of the seedlings of each line under different stress treatments after two weeks of growth. The root length of *BcLEA73-OE* was found to be significantly longer than that of WT (Figure 12B–D,F–H).

### 3.9. Effect of Drought Stress on Seedling Survival Rate

The overexpression lines of transgenic *Arabidopsis* OE#1, OE#2, and OE#3 were selected for drought treatment together with WT. After 10 days of drought treatment, it was observed that the WT plants were more severely damaged than *BcLEA73-O*E. Specifically, the WT plants had more yellow and dry leaves, and more plants had died. After resuming watering for three days, the growth status of most *BcLEA73-OE* plants was restored, while almost all WT plants were dead (Figure 13).

### 3.10. Effects of Salt Stress on the Physiological Indices of the Seedlings

The results showed that, after salt stress treatment, the production rate of superoxide anion in the WT plants increased significantly and was significantly higher than that of *BcLEA73-OE* (Figure 14A). The conductivity was significantly higher than that before treatment, and the conductivity of *BcLEA73-OE* was significantly lower than that of WT (Figure 14B). Under cold stress, the H_2_O_2_ content of *BcLEA73-OE* was significantly lower than that of the WT plants (Figure 14C). The total antioxidant capacity of the *Arabidopsis* plants was enhanced, and the total antioxidant capacity of *BcLEA73-OE* was significantly higher than that of the WT plants (Figure 14D).

### 3.11. The Expression of BcLEA73 in Arabidopsis under Low-Temperature Treatment

We selected *BcLEA73-OE*#3, which had the highest relative expression level, for low-temperature treatment. Its expression level showed an upward trend with the extension of low-temperature treatment time. The expression level peaked at 24 h, which was approximately 2.3-fold higher than that at 0 h (Figure 3, Figure 4, Figure 5, Figure 6, Figure 7, Figure 8, Figure 9, Figure 10, Figure 11, Figure 12, Figure 13, Figure 14 and Figure 15).

### 3.12. Determination of Related Physiological Indices under Low-Temperature Treatment

The related physiological index changes differed significantly under low-temperature treatment. The O_2_^−^ production rate in WT was increased, which was significantly higher than that of *BcLEA73-OE*#3 (Figure 16A). In addition, the H_2_O_2_ content of *BcLEA73-OE*#3 was significantly lower than that of WT (Figure 16B). The total antioxidant capacity of WT and *BcLEA73-OE*#3 was enhanced with the extension of treatment time (Figure 16C).

## 4. Discussion

With the publication of genome data of a large number of species, genome analysis has become an important means to study gene structure, to analyze gene function and evolutionary relationships [16,17,18], and to possibly analyze and identify LEA gene families in the whole genome. Since Wucai will inevitably suffer from abiotic stress in its life cycle, it is an important goal of vegetable breeding to improve the ability of Wucai to resist abiotic stress and to cultivate new varieties of stress resistance. LEA proteins are usually related to protective functions and can help plants resist environmental adversity. The members of the LEA gene family have been identified in many species, though none have been reported in Wucai. Identification and analysis of the gene family members of reported species have indicated differences in the numbers, gene structures, and gene functions of the LEA gene family members across various species [19,20,21,22,23]. In this study, 82 *BrLEA* gene members were identified by analyzing the genome data of Chinese cabbage. Among them, *BrLEA73* belongs to the LEA_6 subfamily, for which there is less research available. Research on *Caragana CkLEA6_1* found that the expression level of this gene changed significantly under low temperatures [24], which is consistent with the results of this study. This indicates that *BrLEA73* responds to low temperature stress in plants and enhances plant tolerance.

In addition, research has indicated that the LEA_2 subfamily genes are the most numerous in higher plants. We also obtained similar results concerning *BrLEA*. In addition, in wheat, the number of members of the LEA_2 subfamily is more than half of the total number of members [21], and, in rye, the number of members of the LEA_2 subfamily accounts for about half of the total [25]. There are also differences in the number of members in each subfamily of different species. The number of members of the LEA_4 subfamily is the largest in cabbage [26], while the Dehydrin subfamily has the largest number of members in rice. The members of the same subfamily have similar gene structures, and there are great differences in gene structures among members of different subfamilies. It can be speculated that the same subfamily genes may have functional similarities, which offers scope for predicting related genes and functional analysis. A large number of studies have found that the LEA protein family in higher plants may be larger and more complex than those currently reported, especially for LEA_2 subfamily proteins [27].

Analysis of the physical and chemical properties of the *BrLEA* gene family members indicated that most LEA proteins are hydrophilic and, therefore, protect plants from dehydration. Many studies have confirmed that genes related to the stress response usually contain fewer introns [28,29,30,31]. Fewer introns contribute to the transcriptional regulation of the LEA gene under plant resistance to stress. This study found that all LEA_6 subfamily members contained 0 introns, which may explain why the *BcLEA* genes of Wucai are rapidly induced or inhibited under drought, salt, and osmotic stress. Similar results were found in Gramineae [32,33].

Chromosomal mapping analysis showed that the 82 *BrLEA* genes were unevenly distributed on 10 chromosomes, and there were three pairs of tandem repeats. Similar results have also been reported for other species, such as the identification of 112 LEA genes in rye, which includes 19 pairs of replication fragments and 12 pairs of tandem repeats [25]. All tandemly replicated genes have undergone intense purification selection. It is speculated that purification selection may play a key role in maintaining the long-term stability of the *BcLEA* biological structure.

The *cis*-acting element is the binding site of transcription factors that activates or prevents gene transcription, and it has been predicted and analyzed for the *BrLEA* gene promoter. The BrLEA gene family members have a variety of *cis*-acting elements in the promoter region, which respond to hormones and abiotic stresses. This study mainly screened the elements related to the stress response, and we speculated that the expression of LEA proteins was related to multiple complex regulatory mechanisms. In this study, most *BrLEA* genes contained elements responding to ABRE, DRE, and LTRE, and the promoter *cis-*acting elements were also confirmed in this study. It is speculated that these genes may play a protective role in related stresses and that the same stress has different induction degrees for different genes, which may enhance the resistance of Wucai to various abiotic stresses.

The results showed that there were significant differences in the expression of the *BcLEA* gene family members in various tissues, among which the expression level of *BcLEA73* was the highest in the leaves. Similar results have also been reported in *Ipomoea pes-caprae* [34] and *Setaria italica* [35], suggesting that LEA genes may have different functions in various tissues during the growth and development of Wucai. In addition, the LEA protein not only highly enriched in late embryonic development, but it is also highly expressed when plants are under stress. Salt, drought, low temperature, ABA, and other treatments can induce greater expression of LEA genes [35,36,37,38]. After low-temperature treatment, the expression of the *BcLEA* gene was analyzed by qRT-PCR. It was found that the expression level of the *BcLEA* gene was significantly upregulated under low-temperature treatment, and the relative expression level of the *BcLEA* gene increased significantly with the increase in low-temperature treatment time. The relative expression level of the *BcLEA73* gene in the leaves, petioles, and other tissues increased, indicating that the *BcLEA73* gene was significantly induced under low temperatures. It can be speculated that the *BcLEA73* gene plays an important role in the process of resistance to low-temperature stress.

To further understand how *BcLEA73* affects the response of plants to abiotic stress, the pCAMBIA1305-35s::*BcLEA73*-nFLAG overexpression vector was constructed. Genetic transformation of Arabidopsis thaliana was carried out by the inflorescence infection method, and its function was analyzed under abiotic stress. The results showed that *BcLEA73* overexpression increased the seed germination rate, root length, and seedling survival rate of *Arabidopsis* under different stress treatments. Under osmotic, salt, and low-temperature stress conditions, the root system of the transgenic *Arabidopsis* was longer, indicating that the *BcLEA73* gene may be highly expressed in the root meristem, thereby promoting root elongation and development. Under low-temperature stress, the T-AOC of the overexpression plants was higher than that of WT, and the H_2_O_2_ content and O_2_^−^ production rate of the WT plants was higher than that of the overexpressed plants. In addition, the expression level of *BcLEA73-OE* increased significantly. In the salt-treated overexpression plants, the REL, H_2_O_2_ content, and O_2_^−^ production rate decreased, and T-AOC increased. However, WT plants accumulated more H_2_O_2_ and O_2_^−^. These changes in physiological indices showed that the overexpression of *BcLEA73* increased antioxidant activity, scavenged free radicals, reduced the accumulation of ROS and cell damage, and it also improved stress resistance [39,40]. Furthermore, the REL of the transgenic plants was lower than that of the control plants. Since electrolyte leakage would increase after membrane damage, these results show that the *BcLEA73* gene can protect the membrane from oxidative damage [41].

## 5. Conclusions

The experimental data further support that *BcLEA73* plays a protective role in enhancing the drought resistance, salt tolerance, low-temperature stress tolerance, and osmotic stress tolerance of *Arabidopsis.* A large number of experimental results show that the overexpression of LEA genes can improve plant resistance to abiotic stress. To the best of our knowledge, this experiment is the first to prove that the overexpression of the *BcLEA73* gene can improve the resistance of transgenic *Arabidopsis* to abiotic stress. These findings not only provide further information for the identification of plant LEA gene family members and related functional analysis, but they also lay a solid foundation for the application of the *BcLEA73* gene in improving plant abiotic stress resistance.

## Figures and Tables

**Figure 1 genes-14-00415-f001:**
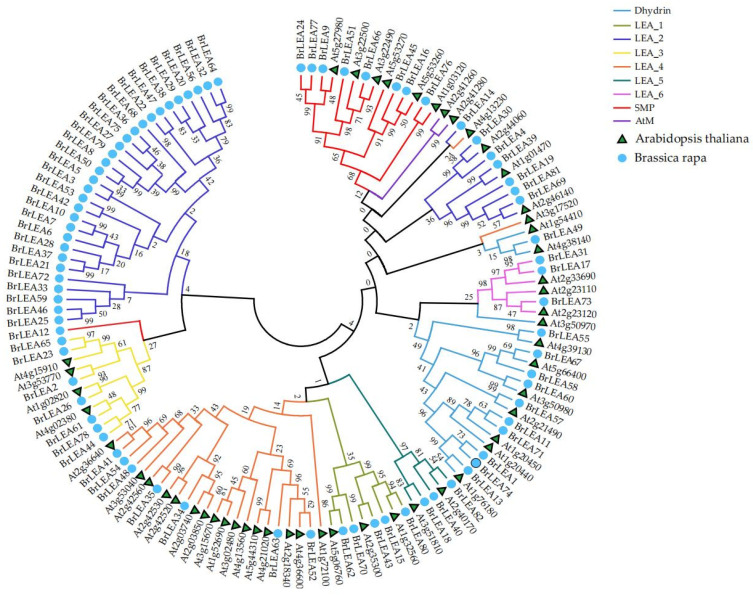
The phylogenetic tree relating to 51 *Arabidopsis AtLEA* [15] and 82 *BrLEA* gene family members in Chinese cabbage. The neighbor-joining phylogenetic tree of the *BrLEA* gene family is represented.

**Figure 2 genes-14-00415-f002:**
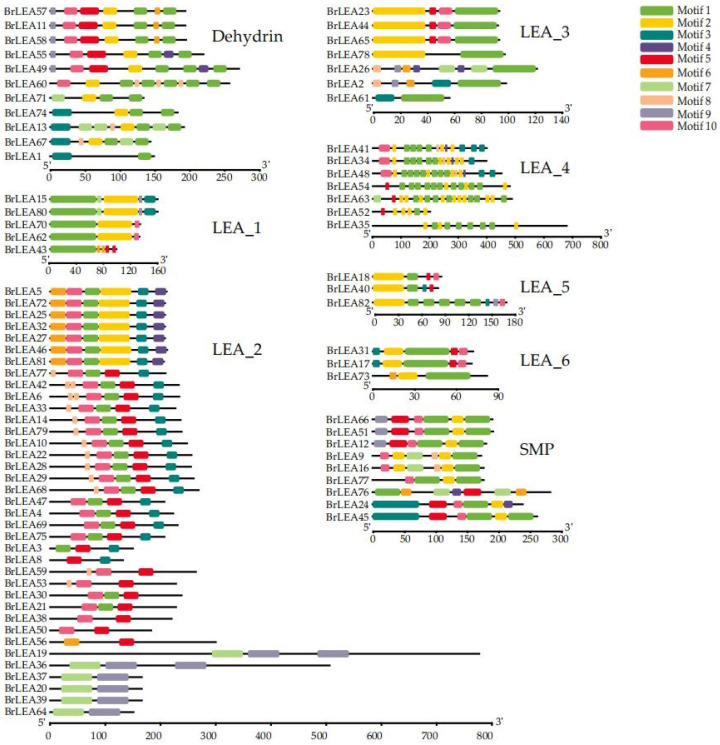
Analysis of the conserved motifs of the *BrLEA* gene family.

**Figure 3 genes-14-00415-f003:**
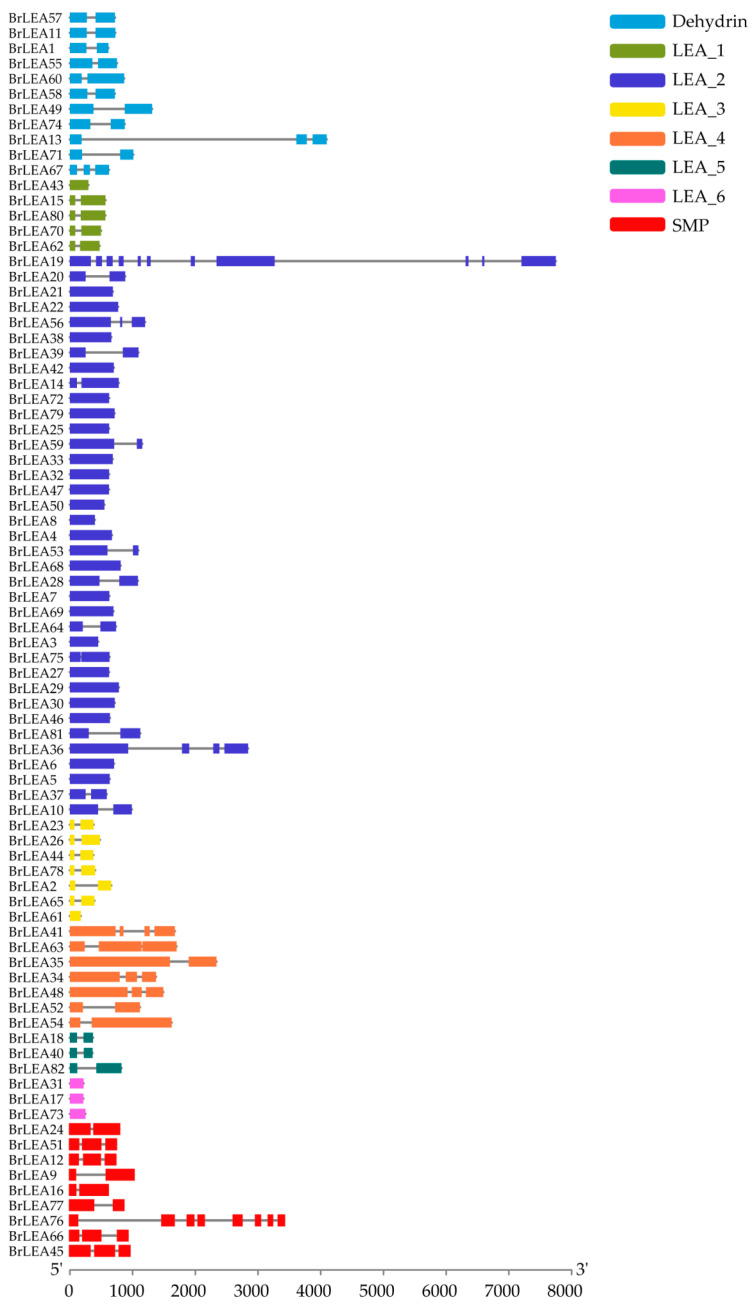
Exon–intron structures of the *BrLEA* gene family.

**Figure 4 genes-14-00415-f004:**
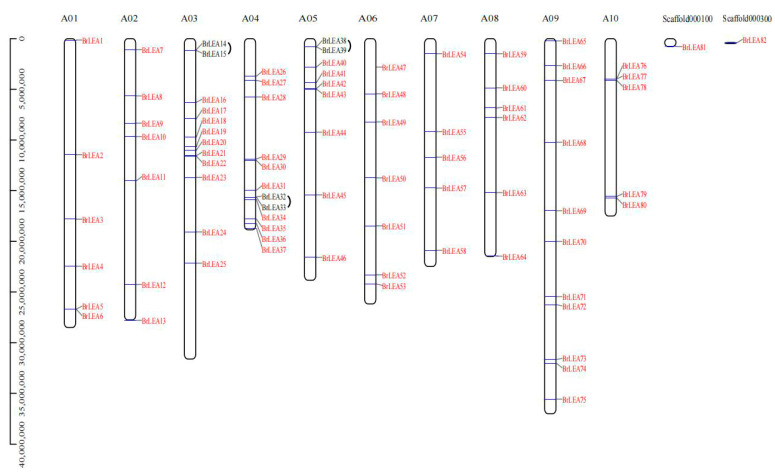
Chromosome localization of the *BrLEA* gene family.

**Figure 5 genes-14-00415-f005:**
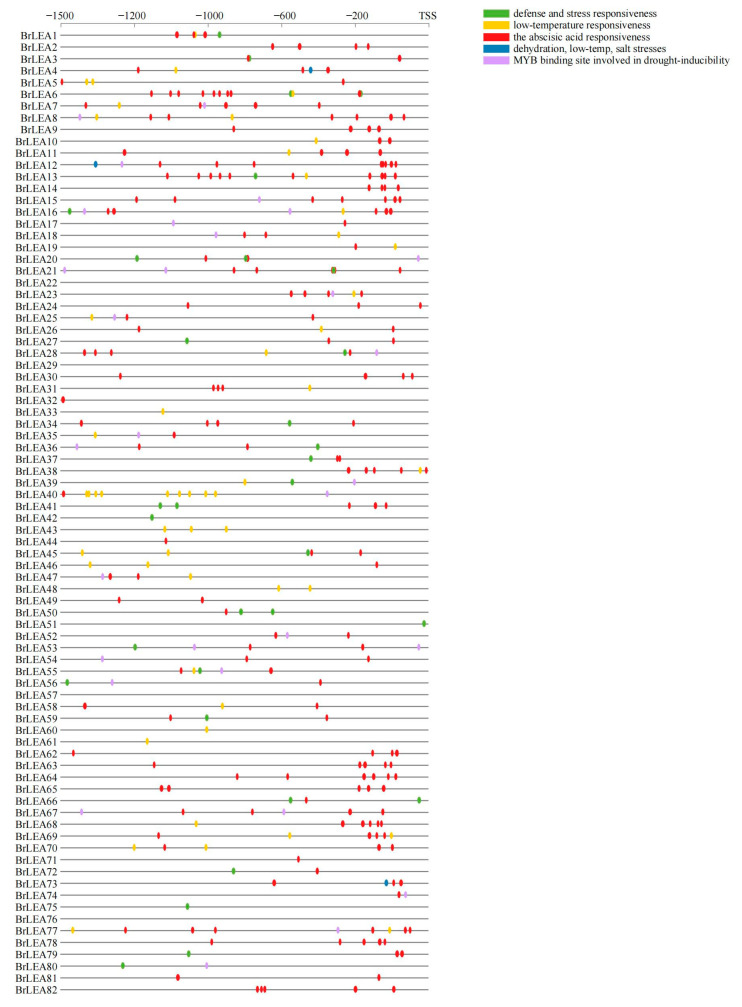
Analysis of *cis*-acting elements on the promoter region of the BrLEA gene family.

**Figure 6 genes-14-00415-f006:**
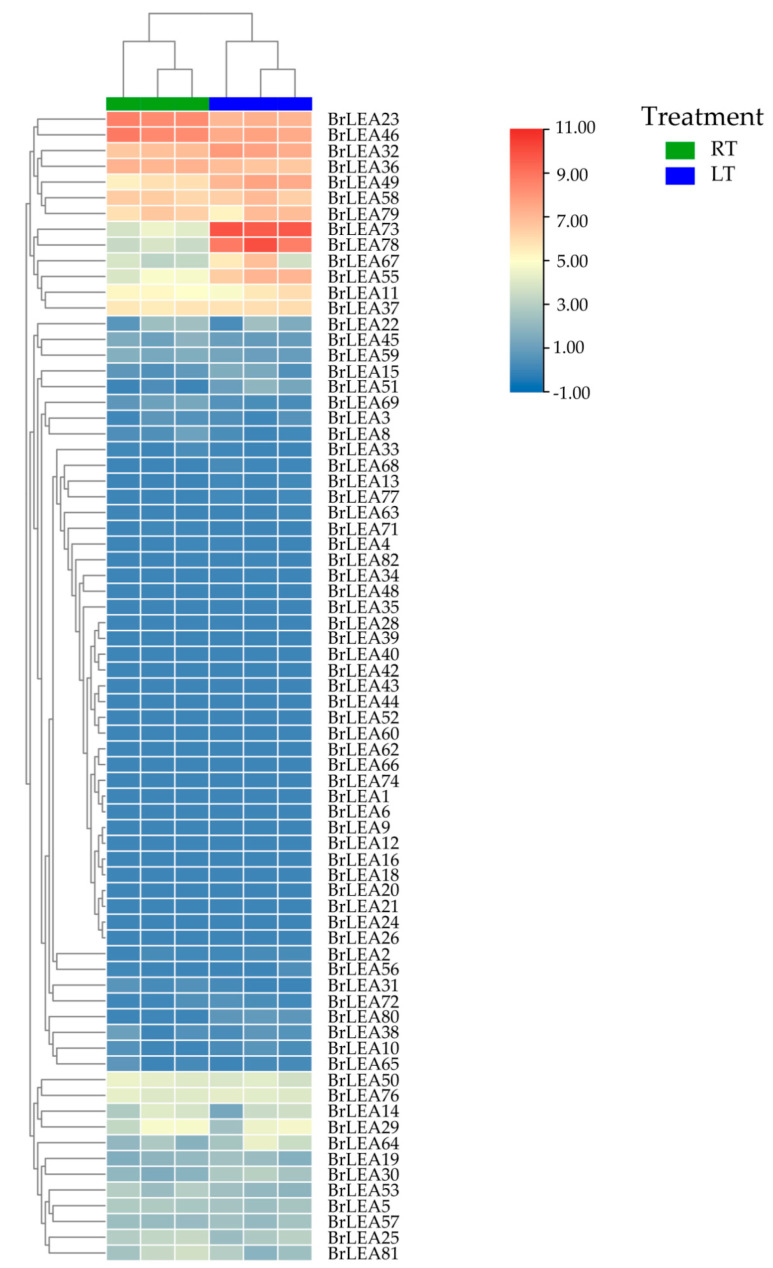
Transcriptome heatmap of regular and low-temperature treatments.

**Figure 7 genes-14-00415-f007:**
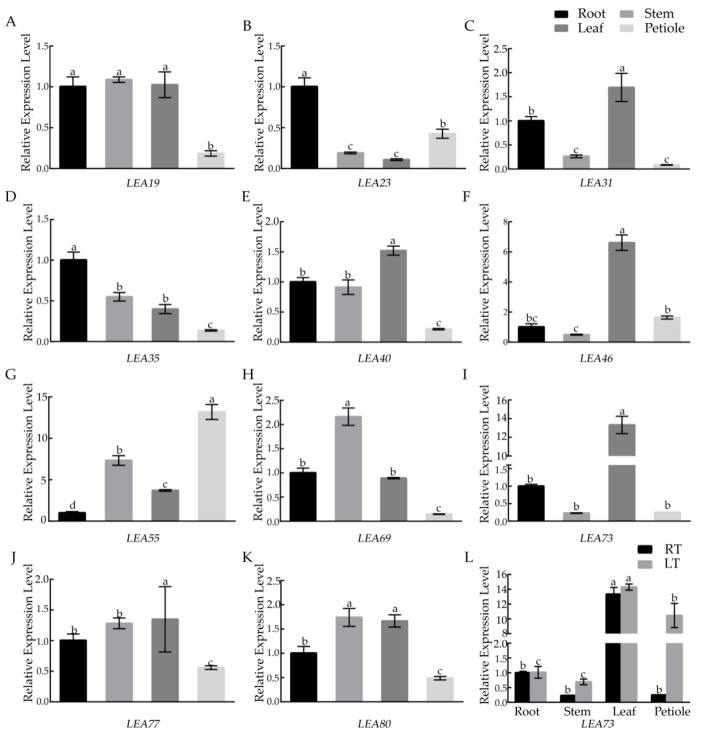
Expression of *BcLEA* genes in different tissues in Wucai. Figure (**A**–**K**) show the relative expression of *BcLEAs* in different organs in Wucai. The relative expression of *BcLEA73* in the roots, stems, leaves, and petioles at regular temperature and low temperature is indicated in (**L**). Bars with different letters “a, b, c, d” above the columns indicate significant differences (*p* < 0.05, Duncan’s range test) on a given day of treatment. Relative gene expression levels were calculated using the 2^−ΔΔCt^ method.

**Figure 8 genes-14-00415-f008:**
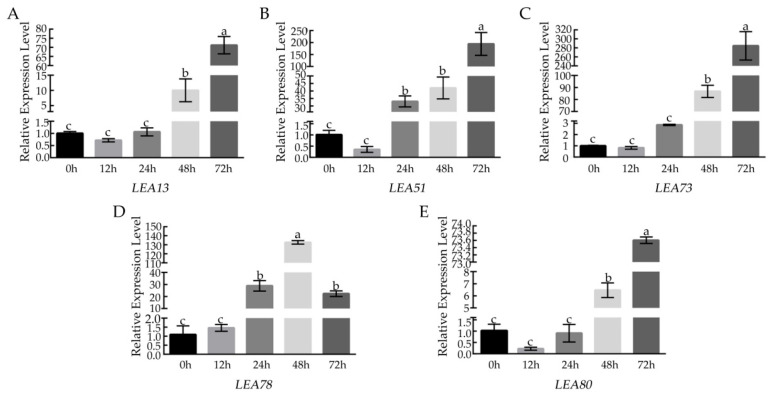
Relative expression level of *BcLEAs* under low-temperature stress. (**A**) Relative expression of *BcLEA13* level under low-temperature treatment. (**B**) Relative expression level of *BcLEA51* under low-temperature treatment. (**C**) Relative expression level of *BcLEA73* under low-temperature treatment. (**D**) Relative expression level of *BcLEA78* under low-temperature treatment. (**E**) Relative expression level of *BcLEA80* under low-temperature treatment. Bars with different letters “a, b, c” above the columns indicate significant differences (*p* < 0.05, Duncan’s range test) on a given day of treatment. Relative gene expression levels were calculated using the 2^−ΔΔCT^ method.

**Figure 9 genes-14-00415-f009:**
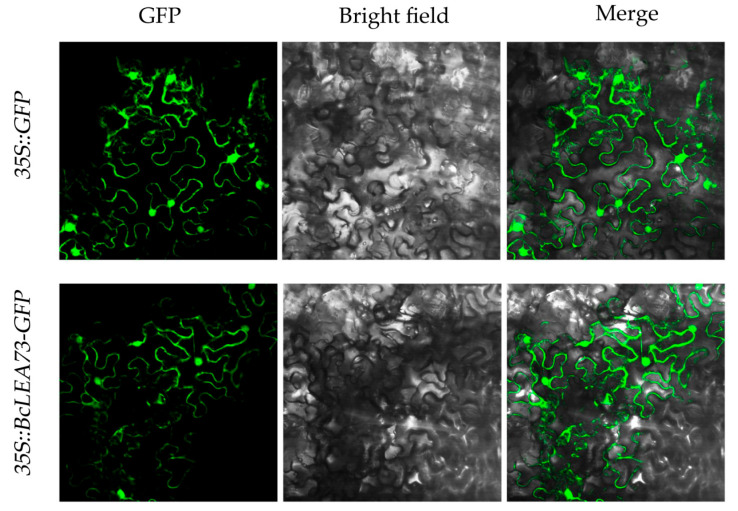
Subcellular localization analysis of *BcLEA73* in Wucai.

**Figure 10 genes-14-00415-f010:**
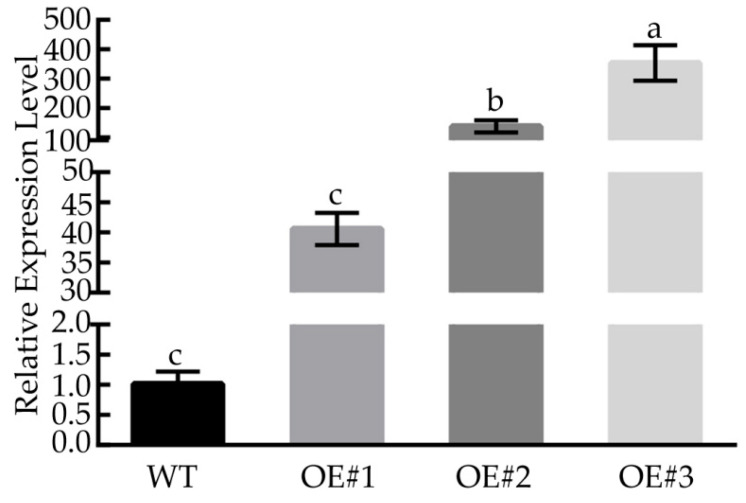
Relative expression level detection of transgenic *Arabidopsis*-resistant plants. Bars with different letters “a, b, c” above the columns indicate significant differences (*p* < 0.05, Duncan’s range test) on a given day of treatment. Relative gene expression levels were calculated using the 2^−ΔΔCT^ method.

**Figure 11 genes-14-00415-f011:**
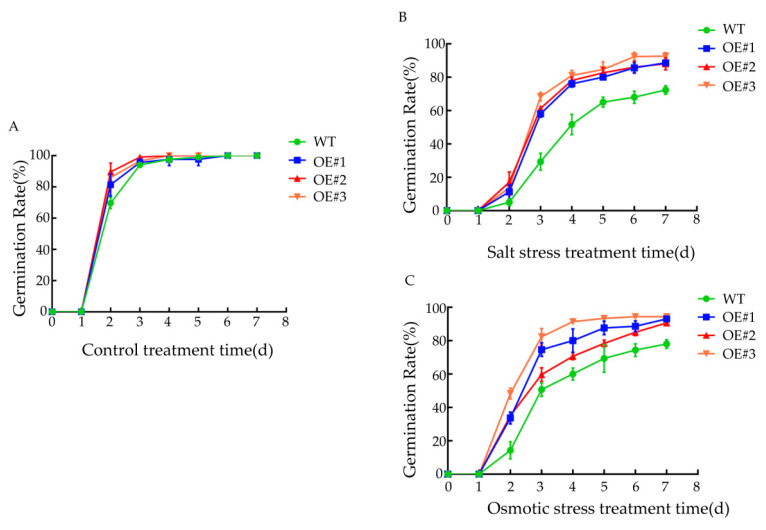
Effects of different stresses on the seed germination rate of transgenic *Arabidopsis thaliana*. (**A**) Half MS Petri dish. (**B**) Half MS Petri dish containing 150 mM NaCl treatment. (**C**) Half MS Petri dish containing 250 mM mannitol treatment.

**Figure 12 genes-14-00415-f012:**
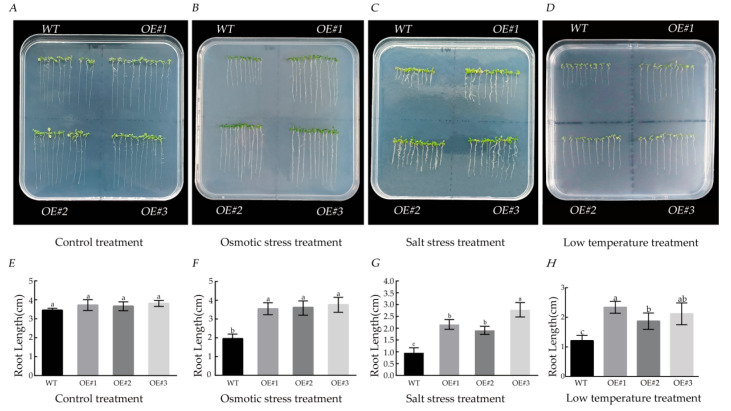
Effects of different stress treatments on the root length of *Arabidopsis thaliana* (**A**,**E**) half MS Petri dish. (**B**,**F**)—Half MS Petri dish containing 250 mM mannitol treatment. (**C**,**G**)—Half MS petri dish containing 150 mM NaCl treatment. (**D**,**H**)—Half MS petri dish treated at a low temperature of 4 °C. Bars with different letters “a, b, c” above the columns indicate significant differences (*p* < 0.05, Duncan’s range test) on a given day of treatment.

**Figure 13 genes-14-00415-f013:**
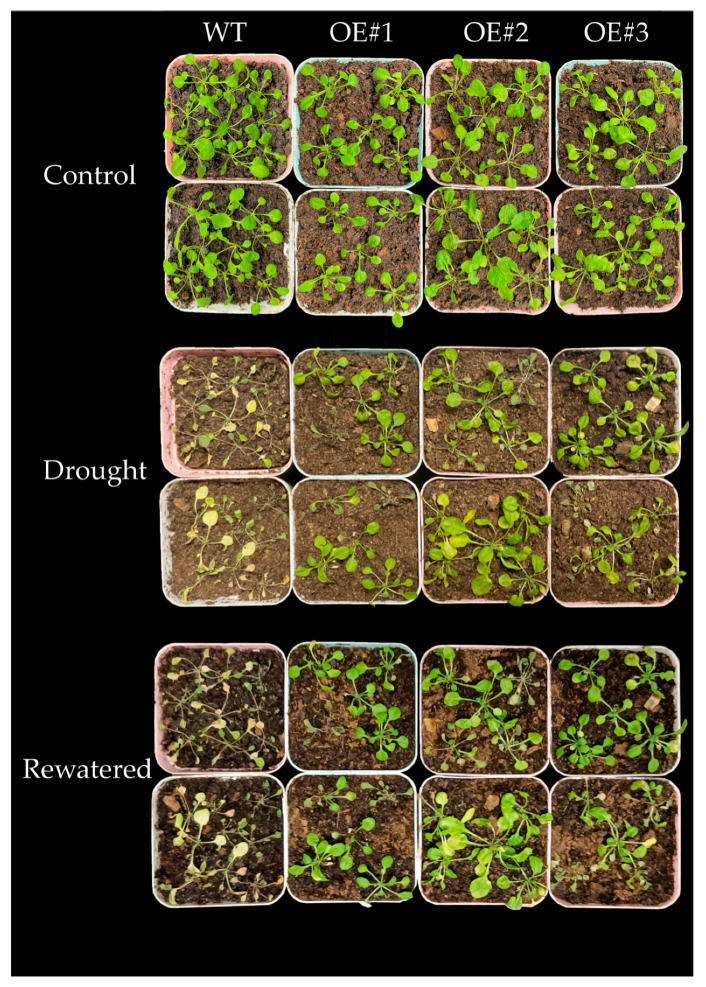
Effects of drought stress on the growth of *Arabidopsis thaliana*.

**Figure 14 genes-14-00415-f014:**
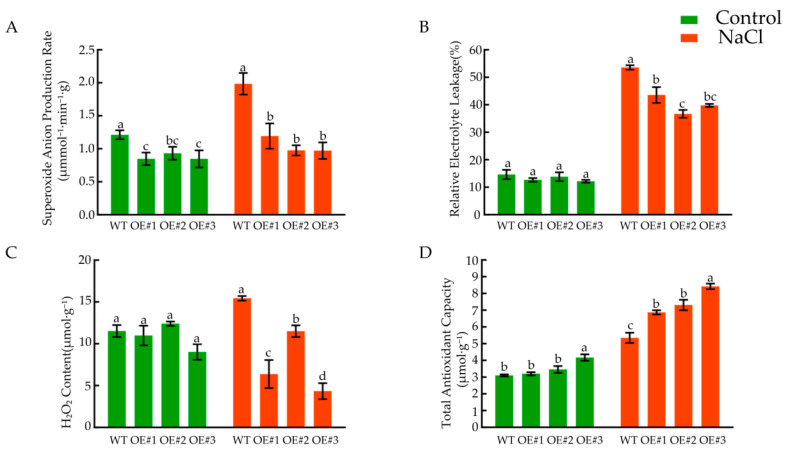
Effects of salt stress on the physiological indices of the seedlings. (**A**) Superoxide anion production rate. (**B**) Relative electrolyte leakage. (**C**) H_2_O_2_ content. (**D**) Total antioxidant capacity. Bars with different letters “a, b, c, d” above the columns indicate significant differences (*p* < 0.05, Duncan’s range test) on a given day of treatment.

**Figure 15 genes-14-00415-f015:**
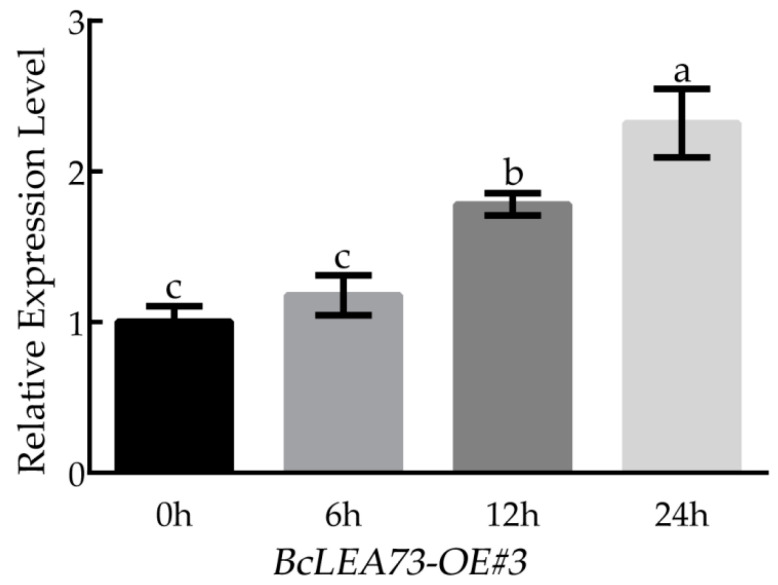
Relative expression level of *BcLEA73-OE#3* under low-temperature treatment. Bars with different letters “a, b, c” above the columns indicate significant differences (*p* < 0.05, Duncan’s range test) on a given day of treatment. Relative gene expression levels were calculated using the 2^−ΔΔCT^ method.

**Figure 16 genes-14-00415-f016:**
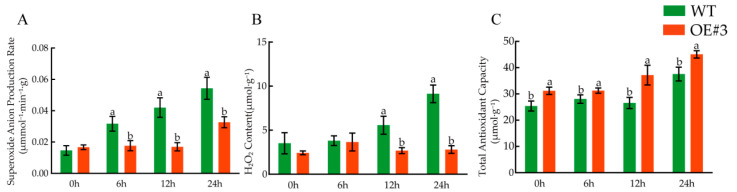
Determination of related physiological indices under low-temperature treatment. (**A**) Superoxide anion production rate. (**B**) H_2_O_2_ content. (**C**) Total antioxidant capacity. Bars with different letters “a, b” above the columns indicate significant differences (*p* < 0.05, Duncan’s range test) on a given day of treatment.

**Table 1 genes-14-00415-t001:** *BcLEA* fluorescence quantitative primer sequence of Wucai.

Primer Name	Primer Sequence (5’—3’)
*RT-BcActin* (F)	TGGGTTTGCTGGTGACGAT
*RT-BcActin* (R)	TGCCTAGGACGACCAACAATACT
*RT-AtActin* (F)	GACCTTTAACTCTCCCGCTA
*RT-AtActin* (R)	GGAAGAGAGAAACCCTCGTA
*RT-BrLEA13* (F)	CGAAGGATACGGGACAGGAAC
*RT-BrLEA13* (R)	GTGAAGCATTCCTCCCAAGCC
*RT-BrLEA51* (F)	AATTACCGTCAGACAAACCAG
*RT-BrLEA51* (R)	ACTCCGGTTGGGTAAGTAGTG
*RT-BrLEA73* (F)	GTGGACGAATCTGGTAGCTTG
*RT-BrLEA73* (R)	TAGATCCACCACCGAGGCCAG
*RT-BrLEA78* (F)	TCTCCAACGCCATCTACAGAC
*RT-BrLEA78*(R)	CTCGCTTGACTCTTCCCCAAC
*RT-BrLEA80* (F)	TGGAGGAGAAGGCTGAGAAGA
*RT-BrLEA80* (R)	TCCACTGGCTTCTTTCATGAC

**Table 2 genes-14-00415-t002:** The primer sequences for PCR amplification.

Primer Name	Primer Sequence (5’—3’)
*BrLEA73*(F)	ATGGAGGCAGAGAAGACAC
*BrLEA73*(R)	TCAAGGAGCTTTCTCAGCAGT

## Data Availability

The raw RNA-Seq data used in this study have been deposited in the Nation Center for Biotechnology Information (NCBI) Sequence Read Archive (SRA) database under the accession number PRJNA735896 “https://www.ncbi.nlm.nih.gov/bioproject/PRJNA735896 (accessed on 6 July 2021)”.

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
