# Peer review of "Identification of the *BcLEA* Gene Family and Functional Analysis of the *BcLEA73* Gene in Wucai (*Brassica campestris* L.)"

_genes, 2023, doi:10.3390/genes14020415_

Round 1

Reviewer 1 Report

This article presented Identification of the BcLEA Gene Family and Functional Analysis of the BcLEA73 Gene in Wucai (Brassica campestris L.). This study provides a theoretical basis to explore the relevant functions of the BcLEA gene family members of B. campestris. Before recommending this article for publication, there are some shortcomings for that should be resolve.

The title. Common name of the species should be used or botanical name is enough. Replace Wucai with common name.

In the abstract the authors should discuss which methods and techniques were used in this study.

Also provide specific quantitative results of the TAC, REL, H2O2 and O2

Economic importance of the B. campestris should be discuss in introduction section.

Discuss the significance of genome wide identification and its main techniques in the introduction.

Also discuss how this study is novelty of the study.

 Section 2.3 must be cited with relevant studies.

Recommended references:

https://doi.org/10.1016/j.indcrop.2022.116090, https://doi.org/10.3390/ijms22179175

Add proper figure legend of the figure 3-5. All legends must be well descriptive.

Modify the conclusion by adding future recommendations and impacts of the current study.

Author Response

The title. Common name of the species should be used or botanical name is enough. Replace Wucai with common name.

A:Wucai is our common name, and we have published articles such as Comparative transcriptome analysis reveals that chlorophyll metabolism contributes to leaf color changes in wucai (Brassica campestris L.) in response to cold under Wucai.

In the abstract the authors should discuss which methods and techniques were used in this study.

A:Already added in the abstract.

Also provide specific quantitative results of the TAC, REL, H2O2 and O2

A: Sorry. No samples available and have left the school, so this part of the data cannot be added

Economic importance of the B. campestris should be discuss in introduction section.

A:Already added in the last paragraph of the introduction section.

Discuss the significance of genome wide identification and its main techniques in the introduction.

A:Already added in the last paragraph of the introduction section.

Also discuss how this study is novelty of the study.

A:Already added in the last paragraph of the introduction section.

 Section 2.3 must be cited with relevant studies.

A:Modifications have been completed.

Add proper figure legend of the figure 3-5. All legends must be well descriptive.

A: A legend is shown at the top of Figure 3.5. Is it possible that our legend is incorrectly labeled?

Modify the conclusion by adding future recommendations and impacts of the current study.

A: I don't understand the future proposals and implications that you mention. The implications of the study are stated in the conclusion. Can you please tell me how to indicate the future recommendations and implications you are referring to?

Reviewer 2 Report

Overall the science is sound, but not well explained. Much of the study is highly descriptive of the 82 LEA genes in Wucai. This section is of only minor value, and it would greatly improve the manuscript to move many parts of this into a supplement (e.g. the first three tables, figure 3-3 and figure 3-5) as they support little discussion. Two critical flaws in the writing: 1) There is little narrative explaining why LEA73 is selected for further study out of the 82 total genes in the descriptive section, 2) the discussion is little more than a recap of the results with scant interpretations. It would behoove the authors to add interpretations to the discussion.

A critical error in the manuscript is a mishandling of tables and figures. Most are not mentioned in the text. All are numbered incorrectly (for example the first table is labelled “Table 2.” Formatting for continuation of tables is poorly done and incorrect. These must all be corrected. This part overall is sloppy.

The GFP fusion experiment is poorly described and must be improved. It is impossible to determine if the interpretation of GFP results is correct. I cannot find any details on a pCAMBIA-2300 35S:GFP vector. Therefore, one cannot know what form the GFP takes (tetrameric or monomeric). If the GFP is prone to tetramer formation, the results are of little use. Furthermore, the control and supposed fusion protein results are indistinguishable from one another which gives little credibility to the claims of nuclear and plasma membrane localization.

There are a number of minor grammatical errors and unclear text throughout that should be corrected.

·      Line 93: “conservative” should be changed to “conserved”

·      Line 109: “were” should be changed to “are”

·      Line 149: “sexual plant screening” is vague and confusing

·      Line 159: “after some time” is completely vague and unacceptable

·      Line 182: check the kDa numbers, 0.66 kDa is off by an order of magnitude for a 57 aa protein.

·      Lines 186-189: this was an algorithmic prediction; therefore, the proteins were not “located” but rather “predicted to be located.”

·      Line 197: “greatest” should be changed to “most abundant”

·      Line 209: please describe the meaning of the colors in the figure

·      Line 255: what statical method was used?

·      Line 272: please label these motifs somewhere in a figure so that the reader can interpret their significance in the sequence

·      Line 279: “injected” should likely read “infiltrated”

·      Line 288: “slices” are highly unusual sampling methodology for Arabidopsis. Please describe this more thoroughly.

·      Line 404: the end of this sentence is confusing, at best.

·      Line 417: missing the verb “is”

·      Line 420-422 (and throughout): “the BcLEA gene” should be changed to “the BcLEA73 gene.”

Author Response

A critical error in the manuscript is a mishandling of tables and figures. Most are not mentioned in the text. All are numbered incorrectly (for example the first table is labelled “Table 2.” Formatting for continuation of tables is poorly done and incorrect. These must all be corrected. This part overall is sloppy.

A: Completed Modified.

The GFP fusion experiment is poorly described and must be improved. It is impossible to determine if the interpretation of GFP results is correct. I cannot find any details on a pCAMBIA-2300 35S:GFP vector. Therefore, one cannot know what form the GFP takes (tetrameric or monomeric). If the GFP is prone to tetramer formation, the results are of little use. Furthermore, the control and supposed fusion protein results are indistinguishable from one another which gives little credibility to the claims of nuclear and plasma membrane localization.

A: We have modified 2.9. Subcellular localization analysis of the BcLEA73 gene in Wucai. The control and supposed fusion protein results are indistinguishable, probably due to improperly selected sites for photography. Do we need to remove the content in response to this question.

There are a number of minor grammatical errors and unclear text throughout that should be corrected.

A: Completed Modified.

Line 93: “conservative” should be changed to “conserved”

A:Completed Modified.

Line 109: “were” should be changed to “are”

A:Completed Modified.

Line 149: “sexual plant screening” is vague and confusing

A:Completed Modified.

Line 159: “after some time” is completely vague and unacceptable

A:Completed Modified.

Line 182: check the kDa numbers, 0.66 kDa is off by an order of magnitude for a 57 aa protein.

A:Sorry, I can’t find the error what you said.

Lines 186-189: this was an algorithmic prediction; therefore, the proteins were not “located” but rather “predicted to be located.”

A:Completed Modified.

Line 197: “greatest” should be changed to “most abundant”

A:Sorry, there is no 'greatest ' in this article.

Line 209: please describe the meaning of the colors in the figure

A:Different colors represent motifs 1-10

Line 255: what statical method was used?

A: Relative gene expression levels were calculated using the 2-ΔΔCT method and we have published articles such as Comparative transcriptome analysis reveals that chlorophyll metabolism contributes to leaf color changes in wucai (Brassica campestris L.) in response to cold under Wucai.

Line 272: please label these motifs somewhere in a figure so that the reader can interpret their significance in the sequence

A:By consulting the literature, we cannot determine the correct domain name, so we delete it.

Line 279: “injected” should likely read “infiltrated”

A:Completed Modified.

Line 288: “slices” are highly unusual sampling methodology for Arabidopsis. Please describe this more thoroughly.

A:Completed Modified.

Line 404: the end of this sentence is confusing, at best.

A:Completed Modified.

Line 417: missing the verb “is”

A:Sorry, I can not accurately locate the error you said.

Line 420-422 (and throughout): “the BcLEA gene” should be changed to “the BcLEA73 gene.”

A:Sorry, these lines we think are correct, please describe more clearly so that we can modify.
